# How Do Transformers Learn Variable Binding in Symbolic Programs?

Yiwei Wu [1]  Atticus Geiger [1]  Raphaël Millière [2]

## Abstract

Variable binding—the ability to associate variables with values—is fundamental to symbolic computation and cognition. Although classical architectures typically implement variable binding via addressable memory, it is not well understood how modern neural networks lacking built-in binding operations may acquire this capacity. We investigate this by training a Transformer to dereference queried variables in symbolic programs where variables are assigned either numerical constants or other variables. Each program requires following chains of variable assignments up to four steps deep to find the queried value, and also contains irrelevant chains of assignments acting as distractors. Our analysis reveals a developmental trajectory with three distinct phases during training: (1) random prediction of numerical constants, (2) a shallow heuristic prioritizing early variable assignments, and (3) the emergence of a systematic mechanism for dereferencing assignment chains. Using causal interventions, we find that the model learns to exploit the residual stream as an addressable memory space, with specialized attention heads routing information across token positions. This mechanism allows the model to dynamically track variable bindings across layers, resulting in accurate dereferencing. Our results show how Transformer models can learn to implement systematic variable binding without explicit architectural support, bridging connectionist and symbolic approaches.

## 1. Introduction

Variable binding, the ability to associate abstract variables with specific values, is a fundamental operation in compu-

[1]Pr(Ai)$^2$R Group [2]Macquarie University. Correspondence to: Yiwei Wu <redroomsd@gmail.com>, Raphaël Millière <raphael.milliere@mq.edu.au>.

*Proceedings of the 42$^{nd}$ International Conference on Machine Learning*, Vancouver, Canada. PMLR 267, 2025. Copyright 2025 by the author(s).

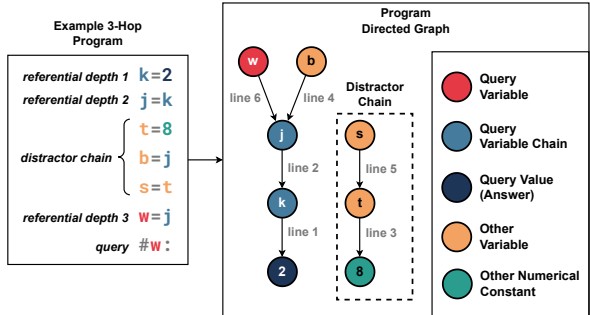

Figure 1. Example 3-Hop Program. While our programs have 17 lines, this example only has 7 lines for illustration. The variable chain of the query variable (w) includes 3 variable assignments or "hops." This program also includes 3 irrelevant variable assignments that act as distractors. An interactive version of this plot for any program can be viewed on 🔗 variablescope.org.

tation and cognition. It enables systems to represent and manipulate structured information by maintaining relationships between abstract roles and their concrete instantiations. In classical computer architectures, variable binding is implemented through addressable read/write memory, where variables serve as addresses pointing to memory locations containing their bound values. This mechanism separates computational machinery from specific values, allowing general-purpose algorithms to operate on arbitrary inputs without requiring dedicated computations for each possible input-output pairing.

The question of how neural networks might implement variable binding, if at all, has been a central point of contention in debates between classicist and connectionist theories of cognitive architecture (Smolensky, 1990; Gallistel & King, 2011; Alhama & Zuidema, 2019). Classicists argue that connectionist models face significant challenges in accounting for the systematic and compositional processing that variable binding enables in symbolic systems. Although some theorists acknowledge that neural networks could in principle exhibit systematic behavior, they suggest that this would merely result from implementing a classical architecture with explicit variable binding operations (Marcus 1998; Marcus et al. 1999; Marcus 2001; cf. Elman 1999; Seidenberg & Elman 1999a;b; Geiger et al. 2023).

The advent of the Transformer architecture provides a new

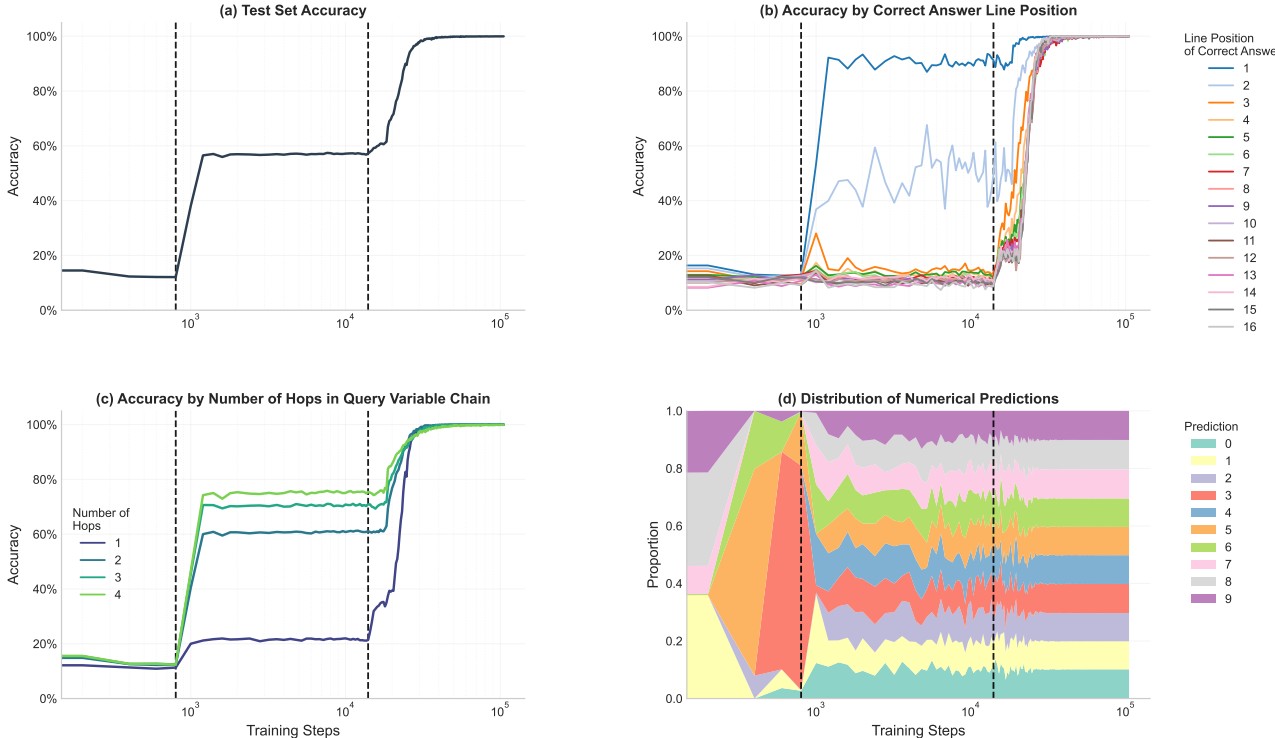

*Figure 2.* Behavioral results showing the learning dynamics of our model across training steps. (a) Overall test set accuracy demonstrates three distinct learning phases, with rapid improvements at and steps. (b) Accuracy breakdown by correct answer line position (1–16) (c) Average accuracy of model checkpoints depending on the number of "hops" of the query variable chain in test set programs. (d) Distribution of model predictions across numerical constants. Vertical dashed lines (at step 800 and 14000) identify the beginning of each major nonlinear transition phase in the developmental trajectory.

avenue for exploring these questions about variable binding in neural networks. Transformers can be seen as implementing a form of read/write memory through their residual stream—the high-dimensional vector space at each token position that acts as a communication channel between layers. This space is addressable through learned linear projections, allowing attention heads to read from and write to specific subspaces (Vaswani et al., 2017). This raises the possibility that Transformers might learn to use their residual stream to implement variable binding operations without explicit architectural support for symbol manipulation.

Recent mechanistic interpretability research has begun to investigate the internal mechanisms underlying variable binding in Transformers. Davies et al. (2023) and Prakash et al. (2024) uncover variable binding circuitry by learning binary masks over model components that mediate variables and values. Additional studies have shown the presence of "binding ID vectors" that associate entities with their attributes, with these vectors occupying a specific "binding subspace" within the activation space (Feng & Steinhardt, 2024b;a; Dai et al., 2024). These findings suggest that variable binding may emerge as a necessary capability for language models to satisfy their training objectives.

In this paper, we investigate the open question of how the capacity for variable binding emerges in neural networks *during training*. We focus on a variable dereferencing task using synthetic programs with a Python-like syntax. These programs consist of 17 lines, where each line (except the last) contains a variable assignment statement of the form `var = const` or `var = var`. The final line contains a query instruction `#var:` that asks the network to output the value bound to the specified variable. Fig. 1 illustrates this task with a simplified 5-line example.

Our analysis uncovers a distinct three-phase developmental trajectory in the model's learning process. The model begins with (1) random predictions of numerical constants, then progresses to (2) shallow heuristics that prioritize early variable assignments, before finally developing (3) a systematic mechanism for variable dereferencing. Through careful causal interventions, we demonstrate that the model learns to use its residual stream as an addressable memory space, with specialized attention heads routing information across token positions to track variable assignments. Notably, we find that the model's final solution builds upon, rather than replaces, its earlier heuristics. This finding challenges traditional narratives suggesting that neural networks abandon

early strategies when they discover a systematic solution to a task. The observed developmental trajectory provides a new perspective on how neural networks can acquire structured reasoning capabilities without explicit symbolic machinery. Our results show that Transformer models can implement fundamental aspects of symbolic computation through learned mechanisms, contributing valuable evidence to ongoing debates about connectionist and symbolic approaches to computation and cognition.

To facilitate transparent and reproducible interpretability research, we developed Variable Scope, an interactive web platform that allows researchers to explore and verify our experimental findings. The platform includes interactive visualizations of the program structure, training checkpoint evaluation, model developmental trajectory, causal intervention experiments, and subspace experiments. This platform builds on previous efforts to present experimental results interactively, such as the Distill Circuits Thread, while providing more granular tools to visualize and analyze the evolution of a neural network over the course of training (Cammarata et al., 2020). Through Variable Scope, we aim to establish a new standard for open and collaborative mechanistic interpretability research: ⌘ variablescope.org.

## 2. Training a Transformer to Run Programs

### 2.1. The Task

Our task requires a model to process programs with variable assignments and determine the final value of a queried variable. Fig. 1 shows an abbreviated 7-line example program. The final line queries variable $w$, which has a value of 2. To determine this value, the model must trace through a chain of variable assignments: $w = j$, $j = k$, and finally $k = 2$. This dereferencing process requires the model to track and retrieve variable bindings throughout the program.

**Referential Depth**  We define *referential depth* as the number of assignment steps needed to reach a numerical value from a queried variable $v$, where for a chain of assignments $v = v_1, v_1 = v_2, ..., v_{n-1} = c$ (with $c$ being a numerical constant), the referential depth is $n$. Our programs contain variable chains with depths ranging from 1 to 4. These chains form a directed graph structure where variables are nodes and assignments are edges. To correctly dereference a queried variable, the model must traverse this graph along the relevant path while ignoring irrelevant branches. For instance, the program in Fig. 1 has a referential depth of 3, as three steps or "hops" are required to trace from the queried variable to its final value ($w \rightarrow j \rightarrow k \rightarrow 2$).

**Distractors**  Our programs include *distractor variable chains* that are either wholly independent from the queried variable chain or branch out from it. A distractor chain is any

sequence of variable assignments that does not contribute to determining the final value of the queried variable but might complicate the task by introducing irrelevant variable relationships. These distractors prevent the network from solving the task through simple pattern matching, instead requiring it to accurately track the relevant variable bindings. For example, in the program shown in Fig. 1, the chain $b \rightarrow j \rightarrow k \rightarrow 2$ represents a distractor that branches from the queried variable's referential chain at variable $j$.

**Program Sampling Procedure**  We generate a dataset of 500,000 programs. Our splits are: training (450,000 programs, 90%), validation (1,000 programs, 0.2%), and testing (49,000 programs, 9.8%). Programs use 26 lowercase letters (a-z) as variables and single-digit numbers (0–9) as values. Every line follows a four-token structure: (1) a variable on the left-hand side, (2) an equality "=", (3) either a variable or a value on the right-hand side, and (4) a newline token.

Each line has a 30% chance of having a numerical constant on the right-hand side of the equality sign, and a 70% chance of having a variable reference, provided that previously defined variables exist. When a variable appears on the right-hand side, multiple variable chains could potentially be extended. We deliberately favor longer chains by sampling with probability proportional to the cube of the chain length. The query variable is selected using the same weighting scheme. Finally, we use rejection sampling to balance the dataset across the four possible referential depths (1–4 hops). This sampling procedure ensures the presence of sufficiently long distractor chains in the data, preventing the model from achieving high performance simply by selecting values associated with the longest chains. Instead, the model must learn to follow the referential paths of each query.

### 2.2. Training a Transformer

We train a Transformer model *from scratch* on our task. Our model is architecturally similar to GPT-2 (Radford et al., 2019), with 37.8M parameters. The model has 12 layers, each with 8 attention heads with a head dimension of 64 and a residual stream dimension of 512. We implement LayerNorm (Ba et al., 2016) before each attention and MLP block, use rotary positional embeddings (RoPE) (Su et al., 2024), apply GELU activations between layers (Hendrycks & Gimpel, 2016), and a dropout rate of 0.1. We do not tie the input and output embedding weights. After training, our model achieves near-perfect performance on the held-out test set (over 99.9% accuracy), showing it has learned a robust mechanism for variable binding and dereferencing.

### 2.3. Phase 1: Predicting Numbers Instead of Letters

In the first phase (training steps 0 to 800), the model performs at approximately chance level, achieving only around

12% accuracy (Fig. 2a). During this phase, the model only learns that the answer should be a numerical value rather than a variable name or special token. This is reflected in the model's prediction distribution, where numerical tokens receive a higher probability mass, though in a notably imbalanced and unstable pattern (Fig. 2d). Despite recognizing that it should output numbers, the model has not yet developed any systematic strategy to identify which particular numerical constant is the correct answer.

## 2.4. Phase 2: Learning Early Line Heuristics

The second phase (training steps 1200 to 14000) begins with a sharp performance jump from 12% to 56% accuracy. During this phase, the model develops two primary prediction strategies or *heuristics*—simplified decision rules that approximate a solution without fully solving the task. These include: (1) a "line-1 heuristic" that selects the numerical constant from the first program line, and (2) a "line-2 heuristic" that selects the numerical constant from the second line (when the right-hand side of that line contains a number). Fig. 2c illustrates this development by showing the model's accuracy over time for programs where the correct answer appears on specific lines. Accuracy for programs with answers on line 1 increases dramatically to over 90%, while accuracy for programs with answers on line 2 reaches approximately 65%. The line-1 heuristic proves particularly effective because our program generation process tends to place the root values of longer reference chains in earlier lines. This is evident in the model's higher Phase 2 accuracy on programs with longer query variable chains (Fig. 2c). For multi-hop programs, the root value—the numerical constant at the end of the chain—frequently appears in the first or second line, making these early-line heuristics surprisingly successful. Conversely, 1-hop programs show slower convergence because their answer (a numerical constant directly assigned to the queried variable) could appear on any line, making these position-based heuristics less reliable.

## 2.5. Phase 3: Systematic Variable Binding

In Phase 3 (training steps 34000 to 105400), we observe another sharp performance transition, with accuracy jumping from 56% to 99%. During this phase, accuracy rapidly improves across all reference depths and distractor configurations, eventually exceeding 99.9% on the test set. The model demonstrates robust performance regardless of query variable chain length (Fig. 2c) or the position of the correct answer within the program (Fig. 2b). This dramatic improvement suggests that the model has acquired a general solution capable of tracing variable chains to any depth while appropriately ignoring distractor assignments.

## 3. Causal Analysis

To understand how the model learns to solve our variable binding task, we use mechanistic interpretability techniques (Cammarata et al., 2020; Saphra & Wiegreffe, 2024; Sharkey et al., 2025) grounded in causal analysis (Cao et al., 2020; 2022; Csordás et al., 2021; Geiger et al., 2021; 2024; Chan et al., 2022; Mueller et al., 2024). These methods allow us to reverse engineer the model's internal mechanisms and provide insights into how neural networks implement symbolic operations. An important feature of our experiment is its developmental perspective (Lovering et al., 2021; Liu et al., 2022; Nanda et al., 2023; Merrill et al., 2023): we track how the network's internal mechanisms evolve throughout training instead of analyzing only the fully trained model.

At the core of our approach is the *interchange intervention* method (Vig et al., 2020; Geiger et al., 2020). This technique involves taking specific components from the model's computation graph and replacing them with values they would have if the model were processing a different, carefully constructed *counterfactual input*—a modified version of the original input that differs in a controlled way. By observing how these interventions affect the model's output, we can identify causal relationships between internal components and model behavior.

**Constructing Counterfactual Inputs**  We use interchange interventions to conduct a *causal tracing* experiment (Meng et al., 2022), allowing us to track how information about specific tokens flows through the model from input to output. While Meng et al. injected noise into activations, our approach instead replaces activations with values from specially constructed counterfactual inputs. For each program in our dataset, we create a counterfactual by first identifying the "root" of the variable chain (the original numerical value that determines the final value of the query variable), and then modifying this root value to a different number. When we intervene by replacing components with their counterfactual values, a successful intervention causes the model to output this new number instead of the original one. This provides a clear signal indicating which model components are responsible for propagating the root value through the network to the final prediction.

### 3.1. Final Checkpoint Analysis

We intervene at two strategic locations in the model architecture. First, we intervene on the residual stream of the Transformer—the block of hidden vectors for each token that serves as input to each layer. Second, we intervene on the outputs of individual attention heads to trace how information moves across token positions. These complementary interventions allow us to pinpoint where and how variable binding information flows through the network.

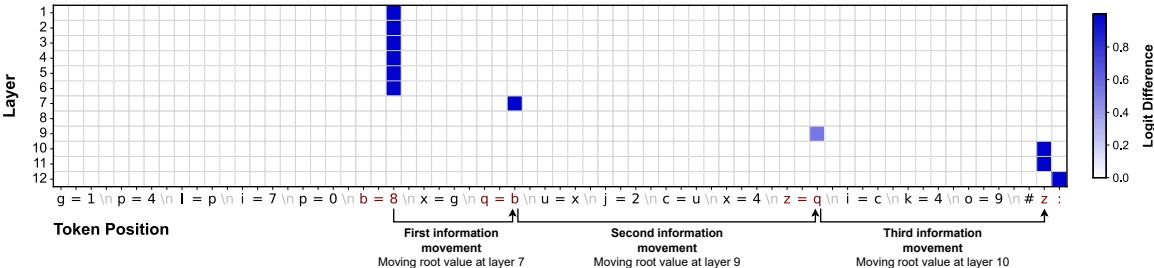

(a): We patch the residual stream incoming to Transformer blocks across all layers and tokens, using a counterfactual input where the root value 8 is replaced with a new number. Logits are computed by running a forward pass with the residual stream at each position replaced by its counterfactual value. Heatmap colors show normalized *logit differences*: the change in pre-softmax activation (logit) of the new number between patched and original runs, divided by the maximum difference. Blue indicates where interventions increase the probability of the new number.

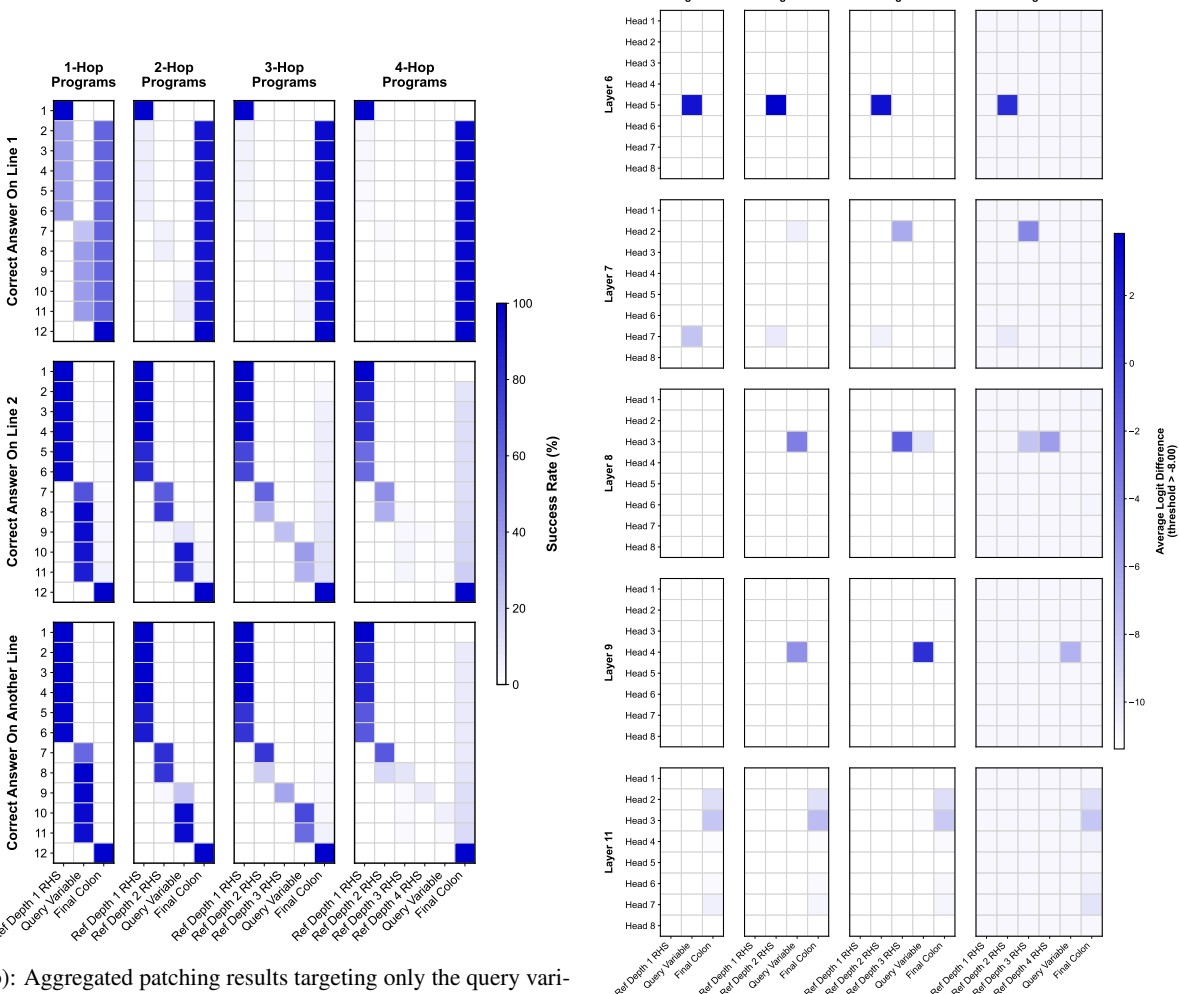

(b): Aggregated patching results targeting only the query variable, the following colon, and the right-hand-side (RHS) tokens at different depths of the query assignment chain (e.g., "8", "b", and "q" in Fig. 3(a)). *Patching success rate* measures how often the intervention causes the model to predict the new number as its top choice, averaged across test programs. A high success rate indicates that the intervened component contains causally relevant information about the output.

(c): Patching results targeting individual attention heads. For each (head, token) position, we replace only that head's contribution to the residual stream with its counterfactual value and compute logits. We analyze only programs where the correct answer appears after line 2. Layers without significant signals are excluded but shown in Appendix K.

*Figure 3.* Patching analysis results on the final model checkpoint. In all experiments, logits are computed by passing the patched hidden states through the unembedding matrix to measure how interventions affect token predictions.

**Tracing a Single Example**  Consider the example program and its pre-residual stream patching results shown in Fig. 3(a). This program contains a query referential chain $(z \rightarrow q \rightarrow b \rightarrow 8)$ formed by three assignment statements: $b = 8$, $q = b$, and $z = q$. Our patching analysis reveals that token locations with high logit difference values across layers correspond precisely to the right-hand side positions of these assignment statements, along with the query variable and colon token in the final query line. This clear pattern of information flow, visible in Fig. 3(b), motivates us to perform systematic pre-residual stream patching experiments on the inputs to each layer.

**Intervening on the Residual Stream**  The attention mechanisms in each layer can move information in the residual stream between different tokens. This path of information flow depends on the specific input provided to the model. When performing interchange interventions on the residual stream, we *dynamically* select token locations rather than intervening at fixed token indices. Our Transformer model uses causal attention, meaning that information can only flow from left to right; consequently, later tokens contain information about earlier tokens. Based on this property, we conduct intervention experiments on tokens that appear on the right-hand side (RHS) of the equality token "=".

Specifically, we intervene along the variable chain that leads to the correct answer, labeling each intervention point according to the referential depth of its line in the chain. For example, in the program shown in Fig. 3(b), we refer to the token "8" as the Ref. Depth 1 RHS token, the second occurrence of "b" as the Ref. Depth 2 RHS token, and so on. We also target the query variable and the colon token following it for intervention. The results in Fig. 3 show two rough patterns. In the first pattern, the number is immediately moved to the colon token at layer 1, which happens for 2-, 3-, and 4-hop programs when the correct answer is on line 1. In the second pattern, the number travels along the query variable chain, which happens in all other graphs to some extent. Notable exceptions are 1-hop programs where the answer is on line 1, which seem split between the two patterns, and 4-hop programs, where the signal is lost entirely after the Ref. Depth 2 RHS token.

These results show that the model develops its general mechanism *on top of existing line-specific heuristics* from Phase 2, overriding them only when needed.

**Intervening on Attention Heads**  We apply the same root value replacement intervention to individual attention heads and present the results in Fig. 3(c). Specifically, we intervene on each head's output as it contributes to the residual stream at specific token positions, allowing us to trace information flow through the network. To isolate the systematic variable binding mechanism, we focus our analysis on pro-

grams where the early-line heuristic (predicting values from initial program lines) fails. For clarity, we exclude layers where all heads show signal below a significance threshold. We use Y.X to denote the Xth head in layer Y.

The attention head patching results reveal a pattern consistent with the information movement observed in the residual stream analysis (Fig. 3(b), third row). For the first hop in the variable reference chain, heads 6.5 and, to a lesser extent, 7.7 play crucial roles. The second and third hops are mediated primarily by heads 7.2, 8.3, and 9.4. In the final processing stage, heads 11.2, 11.3, and 11.7 transfer this information to the output token position.

Interestingly, head 8.3 participates in both the second and third hops of the reference chain. While the first hop processing spans layers 6–7 and the second hop spans layers 7–8, head 8.3 handles both transitions rather than having separate specialized heads for each hop. This is evidence that similar representations are used across hops and token positions for the systematic solution.

## 3.2. Circuit Development Throughout Training

To understand how the systematic mechanism for variable binding emerges over time, we conducted residual stream intervention experiments across different training checkpoints. We track intervention success rates at key (layer, token position) combinations and observe both the persistent influence of early-developed heuristics and the gradual emergence of a systematic variable-tracing mechanism.

We focus on three critical interventions in the network at positions that serve distinct computational roles: (1) Variable value information transferred to the final colon token immediately after layer 1 indicates the operation of line-specific heuristics that bypass multi-step variable tracking. (2) Successful interventions at RHS tokens in layers 6–9 reveal how the model traces variable values by propagating information along reference chains step by step. (3) At the query variable token in layer 10, the model performs the final matching between the query variable and its dereferenced value.

Fig. 4 illustrates the developmental trajectory of our model's variable binding mechanism across programs of different complexity. For 1-hop programs (Fig. 4(a)), we first observe activity at the layer 6 RHS token position (tracing to referential depth 1), followed by the layer 10 query variable position, with patching success rates reaching high levels by training step 60000. For 2-hop programs (Fig. 4(b)), a similar but delayed developmental pattern emerges, with layer 7 and 8 RHS token positions (corresponding to referential depth 2) becoming important.

The developmental pattern becomes more complex for programs with higher referential depths. In 3- and 4-hop programs (Fig. 4(c) and (d)), deeper RHS token positions in

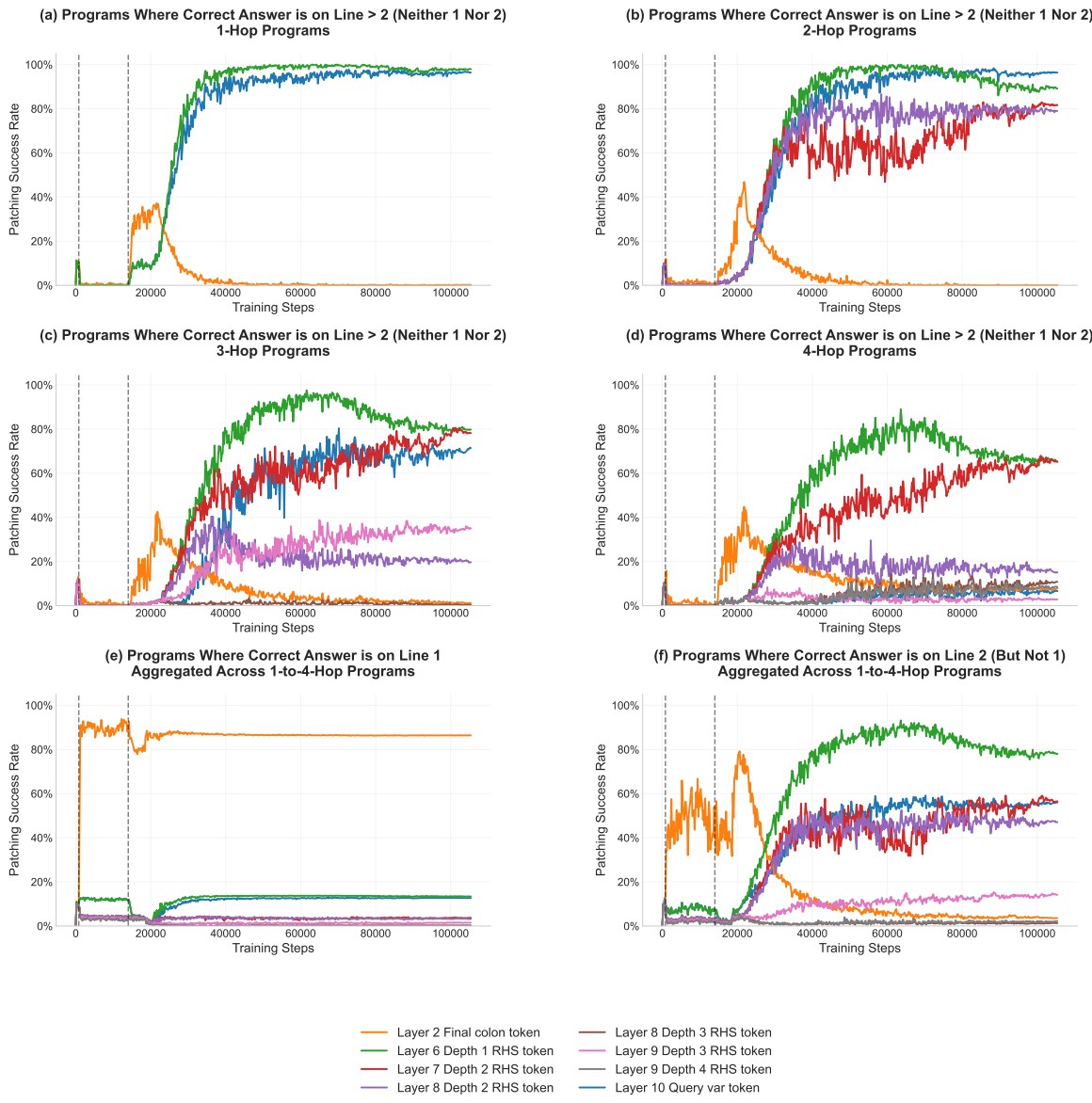

*Figure 4.* Evolution of patching success rates across training steps, revealing the development of line-specific heuristics and a general variable binding mechanism. Vertical dashed lines at steps 800 and 14,000 mark the transitions between major phases identified in our behavioral analysis in Fig. 2. **(a)–(d)** show patching success rates for programs where the correct answer is not on line 1 or 2, separated by the number of hops in the program. We selectively intervene on key (layer, token position) combinations: the query variable token (layer 10), final token (layer 2), and depth-specific RHS tokens (layers 6–9) that our analysis identified as important causal mediation locations for tracking variable bindings where most of the information movement occurs. This highlights the emergence of the general circuit. **(e)** shows patching results aggregated across all programs where the answer is on line 1. The heuristic is consistently used throughout training. **(f)** shows patching results aggregated across all programs where the answer is on line 2. The heuristic is used only in the initial phase.

layers 8 and 9 show both delayed emergence and diminished activation strength. These positions, corresponding to referential depths 3 and 4, exhibit attenuated activation patterns reflecting the increased difficulty of tracking longer reference chains. This systematic correspondence between network layer depth and reference chain length suggests the model develops a specialized circuit that propagates variable binding information through successive layers.

The contrast between Fig. 4(e) and Fig. 4(f) provides valuable insight. Fig. 4(e), depicting programs where the answer appears on line 1, shows that early-learned line-specific heuristics persist throughout training, as evidenced by consistently high success rates (over 80%) when patching the layer 2 final colon token position. In contrast, Fig. 4(f), which focuses on programs where the answer does not appear on line 1, reveals a different pattern: while the heuristic (measured at the layer 2 final colon token) shows initial success, the systematic mechanism (measured at depth 2 RHS locations) develops more slowly compared to Fig. 4(b)–(d).

This contrast reveals that the model builds its variable binding mechanism on top of simpler line-specific heuristics rather than replacing them. The line-specific heuristic (tracked at layer 2's final colon token) remains active throughout training, with the general mechanism (tracked through the RHS token positions) only overriding it when necessary to predict the correct answer. This composite approach enables the model to process simple cases using the heuristic while using more sophisticated algorithms for programs that require them.

### 3.3. Numerical and Variable Subspaces

To investigate how the final model represents and tracks numerical constants and variable names, we hypothesized that these are encoded in separate subspaces within the residual stream. Given that our model uses causal attention, we focus our analysis on the residual streams at token positions corresponding to the right-hand side of each program line in the query chain.

To test this hypothesis, we first aimed to identify these potential subspaces. We applied principal component analysis (PCA) to reduce the 512-dimensional residual stream activations to 256 principal components at the target RHS locations. We performed this analysis on a subset of test samples selected to exclude cases solvable by a line-1 heuristic (where the correct answer is on the first line). This filtering ensures that the observed internal states reflect the systematic mechanism rather than a shallow heuristic.

To isolate the components most relevant to each hypothesized subspace, we trained two linear classifiers with L1 regularization on these 256 principal components. One classifier predicted the numerical constant on the RHS of the

line (targeting the numerical constant subspace). The other predicted the variable name typically found on the LHS (targeting the variable name subspace, probed via the RHS token's residual stream). The L1 penalty encourages sparsity, effectively selecting a small subset of components most predictive for each task. This process yielded 10 principal components for numerical constants and 26 for variable names.

Using these components, we then extracted activations from the residual stream feeding into layer 6 across a subset of programs. To visualize the structure within these subspaces, we projected the vectors in 2D using UMAP (McInnes et al., 2018). We generated visualizations at three distinct training checkpoints (steps 1200, 17600, and 105400) to observe their evolution, as shown in Fig. 5. The UMAP projections demonstrate increasing separation over training. By the final checkpoint, distinct clusters emerge for different numerical values (Fig. 5(a)) and variable identities (Fig. 5(b)), revealing the numerical and variable subspaces.

To validate the causal role of these subspaces, we performed interchange interventions on each one. Unlike earlier interventions, these swapped only the residual stream subspaces spanned by the selected component sets (10 for numerical constants, 26 for variable names) between original and counterfactual programs targeting either the numerical constant or the variable name. These interventions had high success rates: 92.17% for numerical constants and 87.08% for variable names. This provides strong causal evidence that the identified subspaces encode their respective information types, confirming our hypothesis.

## 4. Discussion

Our study shows how neural networks can acquire capabilities traditionally associated with symbolic computation, particularly variable binding and dereferencing. Analysis of the model's learning trajectory reveals that the path to systematic variable binding proceeds through distinct phases with qualitatively different strategies.

Notably, we find that the model's final solution builds upon, rather than replaces, the heuristics learned in earlier phases. This adds nuance to the traditional narrative about "grokking", where models are thought to discard superficial heuristics in favor of more systematic solutions. Instead, our model maintains its early-line heuristics while developing additional mechanisms to handle cases where these heuristics fail, suggesting cumulative learning where sophisticated capabilities emerge by augmenting simpler strategies.

Causal interventions show the model uses its residual stream as addressable memory, with specialized attention heads routing information to track variable assignments. This

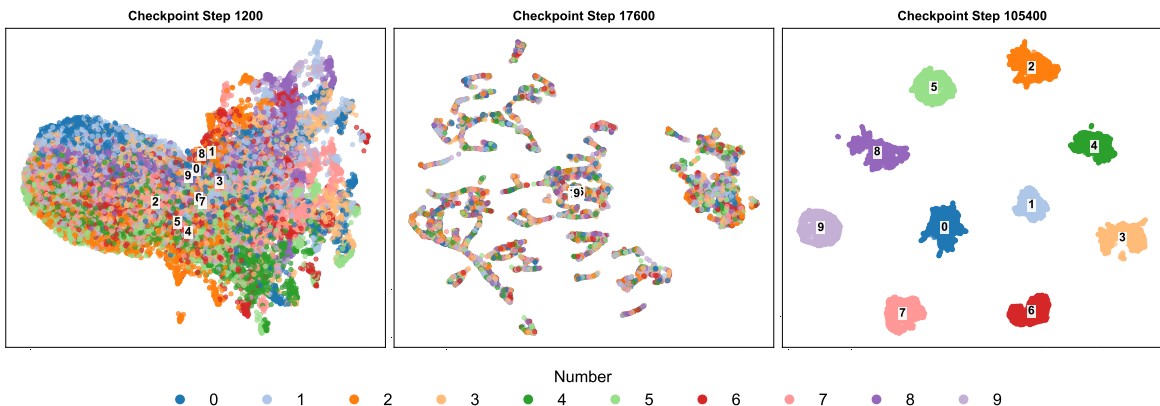

(a): 2D UMAP projection of the 10 principal components identified as predictive of numerical constants. PCA was first applied to RHS token residual streams (from non-heuristic test samples) reducing dimensionality from 512 to 256. An L1-regularized linear classifier then selected these 10 components. The projection visualizes these residual stream activations (from the input to layer 6) across three training checkpoints: steps 1200, 17600, and 105400.

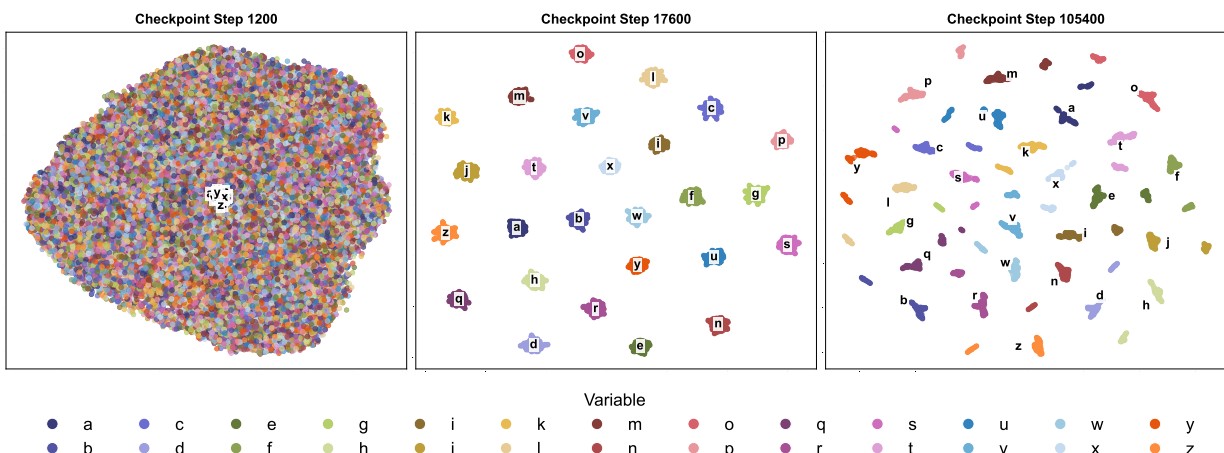

(b): 2D UMAP projection of the 26 principal components identified as predictive of variable names. PCA was first applied to RHS token residual streams (from non-heuristic test samples) reducing dimensionality from 512 to 256. An L1-regularized linear classifier then selected these 26 components. The projection visualizes these residual stream activations (from the input to layer 6) across three training checkpoints: steps 1200, 17600, and 105400.

*Figure 5.* Evolution of 2D UMAP for numerical-constant and variable-name residual stream subspaces (input to layer 6) across training.

provides concrete evidence for how symbolic computation can emerge from continuous vector operations. Importantly, our probing experiments suggest that the model develops an efficient strategy focused on tracking only the information necessary for dereferencing, rather than maintaining complete program state.

These findings have implications for cognitive science and machine learning, showing how symbolic capabilities can emerge from neural architectures without built-in symbolic operations. This also suggests that the development of sophisticated reasoning capabilities might be better supported by training regimes that allow progressive refinement and composition of simpler strategies.

## 5. Conclusion

Our study reveals how Transformers can learn to perform variable binding and dereferencing—a capability traditionally associated with symbolic computation—without built-in symbolic operations. Through mechanistic analysis, we demonstrate that the model implements variable binding by repurposing its residual stream as an addressable memory space. These findings contribute to ongoing debates about connectionist and symbolic approaches to computation. We present an interactive web platform ⚲ variablescope.org to support reproducible research.

## Acknowledgments

This work was in part supported by a grant from Open Philanthropy.

## Impact Statement

This research was conducted with the goal of understanding how language models develop and combine different circuits. By contributing reproducible mechanistic interpretability research and open scientific methodology, this work supports broader efforts to make AI systems more transparent and understandable.

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

## A. Expanded Related Works

**Synthetic Tasks.** Our work expands on previous research using synthetic tasks to investigate how Transformers handle highly structured data, including reference chains. Zhang et al. (2023) showed through their LEGO task that Transformers can develop specialized attention heads for matching identical tokens and processing local operations, suggesting mechanisms for tracking variable relationships. the hashhop benchmark (Magic, 2024) assesses whether models can learn to resolve chains of references in long contexts while avoiding pattern-matching shortcuts. while these works focus on length generalization and path-finding in graphs, our task emphasizes sequential execution order and memory operations, providing a focused test bed for investigating variable binding mechanisms in Transformer architectures.

**Variable Binding in Pretrained Transformers.** Recent work has also made progress in understanding how pretrained Transformer-based language models may implement variable binding mechanisms. Our work builds on and complements these findings by providing a detailed mechanistic account of how variable binding emerges during training.

Davies et al. (2023) developed an automated approach to identify shared variable binding circuitry in LLaMA-13B that retrieves variable values for multiple arithmetic tasks. Using causal mediation experiments with carefully designed desiderata, they localized variable binding to 9 attention heads and one MLP in the final token's residual stream. This suggests that variable binding capabilities can be implemented by a sparse subset of model components working together.

Feng & Steinhardt (2024a) analyzed LM representations and identified a general binding ID mechanism present in every sufficiently large model from the Pythia and LLaMA families. They showed that LMs' internal activations represent binding information by attaching binding ID vectors to corresponding entities and attributes, with these vectors forming a continuous subspace. Through careful causal intervention experiments, they found that binding IDs are used consistently across different tasks and can be transplanted between tasks, suggesting the emergence of a general binding mechanism.

Building on this work, Dai et al. (2024) discovered that LMs encode ordering information in a low-rank subspace that causally determines binding behavior. By using dimensionality reduction techniques, they identified an "Ordering ID" subspace distinct from pure positional information, providing a novel geometric view of how binding is implemented. Their causal intervention experiments showed that editing activations in this subspace could systematically alter which attributes are bound to which entities.

Feng et al. (2024) introduced "propositional probes" to monitor how language models internally represent relationships between entities and their attributes. Using a Hessian-based algorithm, they identified a binding subspace where tokens that should be bound together (like a person and their occupation) have high similarity, while unbound tokens do not. This binding subspace was found to be fairly robust, maintaining accurate representations even when the model's outputs became unfaithful due to biases or adversarial attacks.

Prakash et al. (2024) studied how fine-tuning affects entity tracking mechanisms, finding that fine-tuning enhances existing mechanisms rather than creating new ones. They showed that the same circuit implementing entity tracking in the base model persists in fine-tuned versions with improved performance, suggesting that binding capabilities can be strengthened through targeted training.

Our work complements these findings by providing a detailed analysis of how variable binding emerges during training. While prior work has focused on identifying binding mechanisms in pretrained models, we examine the developmental trajectory through which these mechanisms are learned. Our results show that the model progresses through distinct phases—from random prediction to shallow heuristics to systematic binding—with rapid nonlinear transitions between phases. This developmental perspective provides new insights into how neural networks acquire structured reasoning capabilities.

## B. Program Structure

A sample synthetic program (abbreviated from the full 17-line version) looks like this:

```
1  a = 1
2  b = a
3  d = 2
4  e = d
5  c = b
6  f = e
7  # c :
```

The lines highlighted in green form the reference chain for the queried variable c, while the lines in orange show an independent distractor chain.

The synthetic program structure can be formally described by the following grammar:

$$\text{program} \rightarrow \text{stmt}^{16}\ \text{query}$$
$$\text{stmt} \rightarrow \text{var} = (\text{const} \mid \text{var})$$
$$\text{query} \rightarrow \#\text{var}:$$
$$\text{var} \rightarrow [\text{a-z}]$$
$$\text{const} \rightarrow [\text{0-9}]$$

The design of our synthetic programs intentionally uses a minimal symbolic vocabulary, with only numerical constants (0–9) and single-character variables (a–z). This controlled environment enables precise causal analysis of how variable binding mechanisms emerge during training.

Despite its apparent simplicity, the task demands sophisticated computational capabilities: the model must track multi-step variable dependencies through chains of references while ignoring distractor assignments.

## C. Tokenization Details

We employ a simple character-level tokenizer where each individual character is treated as a single token, including all variable names, numerical constants, and special characters like #, \n and =.

## D. Choice of Sampling Strategy

We carefully selected the program sampling strategy to ensure it would effectively test the model's capacity to track variable assignments. Early experiments with simpler distributions showed that models could solve the task without developing genuine variable binding mechanisms, instead relying on surface-level heuristics.

We found that several distribution characteristics made the task too simple. Programs with fewer distractor chains, uniform sampling of variables, or shorter reference chains allowed models to develop shortcuts. For instance, without a weighted sampling approach, models could simply learn to associate the correct answer with the variable appearing in the longest chain, bypassing the need to track actual variable bindings.

To address this, we implemented a sampling strategy where chains are selected for extension with probability proportional to the chain length cubed. This cubic weighting creates programs with multiple lengthy chains, preventing the model from relying solely on chain length as a heuristic. Additionally, we used rejection sampling to balance our dataset across the four referential depths while maintaining sufficient distractor chains that branch from the main query chain. The resulting distribution effectively forces the model to track specific variable bindings through the program's execution rather than using pattern matching based on surface features.

## E. Training Setup

We train our model from scratch using standard causal language modeling on complete sequences, where each sequence consists of the 17-line program string (16 lines of variable assignments and a final query line) and its expected

result. Importantly, the model has no prior exposure to natural language or programming data, and must learn a mechanism for variable binding and dereferencing purely from our synthetic data. The model is trained for 15 epochs using the AdamW optimizer with $\beta_1 = 0.95$, $\beta_2 = 0.999$, and a base learning rate of $1 \times 10^{-4}$ (Loshchilov & Hutter, 2017) with a batch size of 64 programs.

The learning rate follows a linear decay schedule with warmup. Starting from zero, the learning rate linearly increases to $1 \times 10^{-4}$ over 750 warmup steps, then linearly decays to zero over the remaining training period. For regularization, we applied dropout with a rate of 0.1 and a weight decay coefficient of $1 \times 10^{-4}$.

## F. Training Accuracy Across Multiple Random Seeds

We examined the stability of the observed learning phenomena by performing multiple training runs, each initialized with a different random seed (specifically, seeds 42, 256, 416, 512, 1024, and 3407). Seed 42 corresponds to the original training run analyzed earlier in the main text. The evolution of test accuracy during training for each of these runs is shown in Fig. 6. All runs exhibit a learning dynamic characterized by three distinct phases with similar transition points, consistent with the pattern identified in Fig. 2. This reproducibility strongly suggests that the observed three-phase learning dynamic is a robust property of the training process rather than an artifact of a specific random initialization.

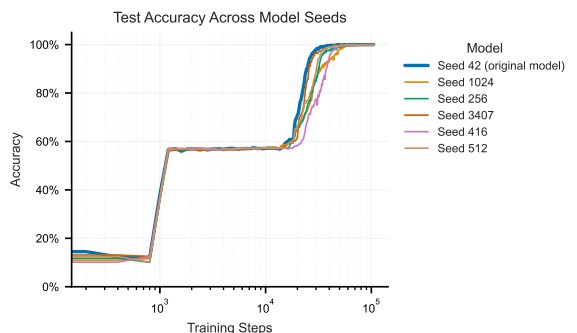

*Figure 6.* Comparison of test set accuracy curves across multiple training runs initiated with different random seeds. All runs exhibit similar three-phase learning dynamics and transition points with slightly different convergence speeds.

## G. Generalization to Unseen Combinations

We explicitly checked for potential memorization by training a model on a subset of the original training data. Specifically, 10% of variable/number combinations were randomly

sampled and excluded from the training set but added to the test set. As illustrated in Fig. 7, the accuracy progression and final performance of this model closely mirror those of the model trained on the complete dataset. This indicates that the model effectively generalizes to novel combinations it was not exposed to during training, confirming its ability for compositional reasoning rather than mere memorization.

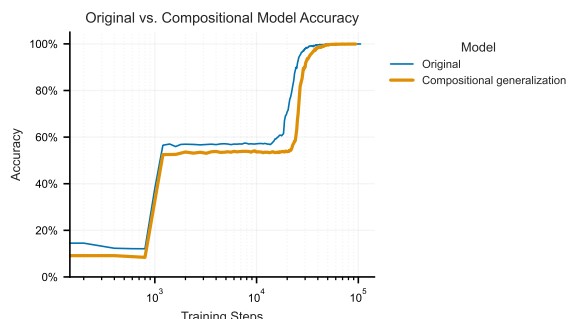

*Figure 7.* Training accuracy comparison for evaluating compositional generalization. The model trained with 10% of variable/number combinations held out achieves comparable test accuracy throughout training to the model trained on the full dataset.

## H. Generalization to Longer Programs

To assess generalization beyond the training distribution, we evaluated the model on programs with lengths varying from 2 to 25 lines, contrasting with the fixed training length of 16 lines. Fig. 8 presents the model accuracy stratified by the line number containing the correct answer. The results indicate robust generalization when the correct answer is located on line 2 or beyond, where high accuracy is maintained across diverse program lengths. However, when the correct answer is on line 1, the model appears to rely on a shallow line-1 heuristic. Consequently, the model exhibits poor generalization for programs substantially different in length from the training regime, with accuracy dropping sharply for lengths below 14 and above 17 lines, since the learned heuristic itself does not generalize well beyond the specific 16-line structure seen during training.

## I. Generalization to Programs with More Hops

We also evaluated the model's ability to generalize to programs with significantly more hops than encountered during training. While trained on programs with a maximum of 4 hops, the model was tested on programs ranging from 1 to 13 hops. The results are shown in fig. 9. For programs where the correct answer lies beyond the first line, the model demonstrates high accuracy across all hop counts tested. This suggests the acquisition of a systematic mech-

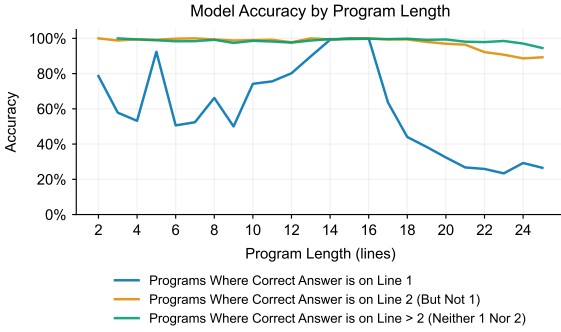

*Figure 8.* Generalization performance across varying program lengths (2–25 lines). Performance is broken down by the line number containing the correct answer. While the model (trained on length 16) generalizes well when the answer is on line 2 or later, accuracy drops significantly for first-line answers in programs shorter or longer than the training length, as the model leans on line-1 heuristic that failed to generalize on program lengths.

anism rather than reliance on memorized patterns that is specific to up to programs with up to 4 hops in the training set.

On the other hand, when the correct answer is located on the first line, the accuracy degrades faster as the number of hops increases. This performance drop aligns with the limitations of a shallow heuristic, similar to the results on program length generalization.

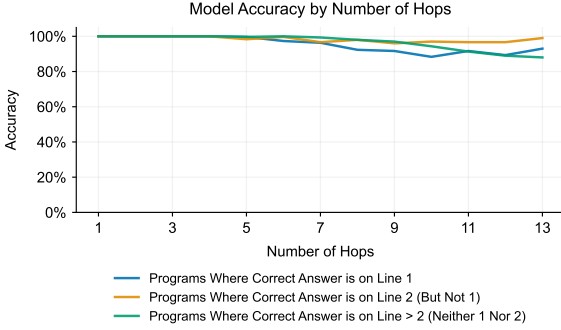

*Figure 9.* Generalization performance across varying program hop counts. The model maintains high accuracy on programs up to 13 hops (far exceeding the maximum of 4 seen during training) when the correct answer is not on lines 1 or 2, indicating a systematic solution. In contrast, accuracy degrades for programs where the answer resides on line 1 or line 2 as hop count increases.

## J. Linear Probing Results

The model's strong performance suggests it might develop structured internal representations of program states, potentially simulating the execution of variable assignments line by line. To investigate this hypothesis, we applied linear probes to the output of each Transformer layer, specifically at newline token positions (\n) where program state updates occur. These probes attempt to extract information about the current program state after each line of code.

For our analysis, we represented program states as $(26, 11)$ tensors—one dimension for each possible variable name (26 lowercase letters) and another for each possible value (digits 0–9 plus nil for unassigned variables). We trained separate linear probes for each layer using the final model checkpoint.

The results presented in Table 1 do not support our hypothesis of explicit program state representation: even the best-performing layer (layer 6) achieves only 30.87% accuracy when predicting variable values (excluding unassigned variables) and merely 8.90% accuracy when predicting complete program states. These poor results suggest the model does not maintain a complete program state in a linearly decodable format at any single vector location.

This finding is particularly significant when contrasted with our successful causal intervention experiments. While linear probing attempted to extract a complete program state from individual vectors, our patching experiments revealed the dynamic flow of specific information through the network. The success of these causal interventions indicates that the model implements variable binding not as static state representations but as a dynamic process of information routing. The model appears to have learned a more efficient strategy that tracks only the relevant variable bindings through specialized attention patterns, rather than representing the entire program state—a hypothesis further explored in our main analysis.

| Layer | State Acc (%) | Var. Acc (Excl. Nil) (%) |
|---|---|---|
| 1 | 7.71 | 21.28 |
| 2 | 8.42 | 25.36 |
| 3 | 8.78 | 28.56 |
| 4 | 8.87 | 28.52 |
| 5 | 8.73 | 29.80 |
| 6 | **8.90** | **30.87** |
| 7 | 8.88 | 30.72 |
| 8 | 8.83 | 30.03 |
| 9 | 8.85 | 29.61 |
| 10 | 8.73 | 28.90 |
| 11 | 8.72 | 28.66 |
| 12 | 8.77 | 28.51 |

*Table 1.* Linear probing accuracy at program state and variable (excluding nil values) levels by layer.

## K. Single Head Activation Patching Results for All Layers

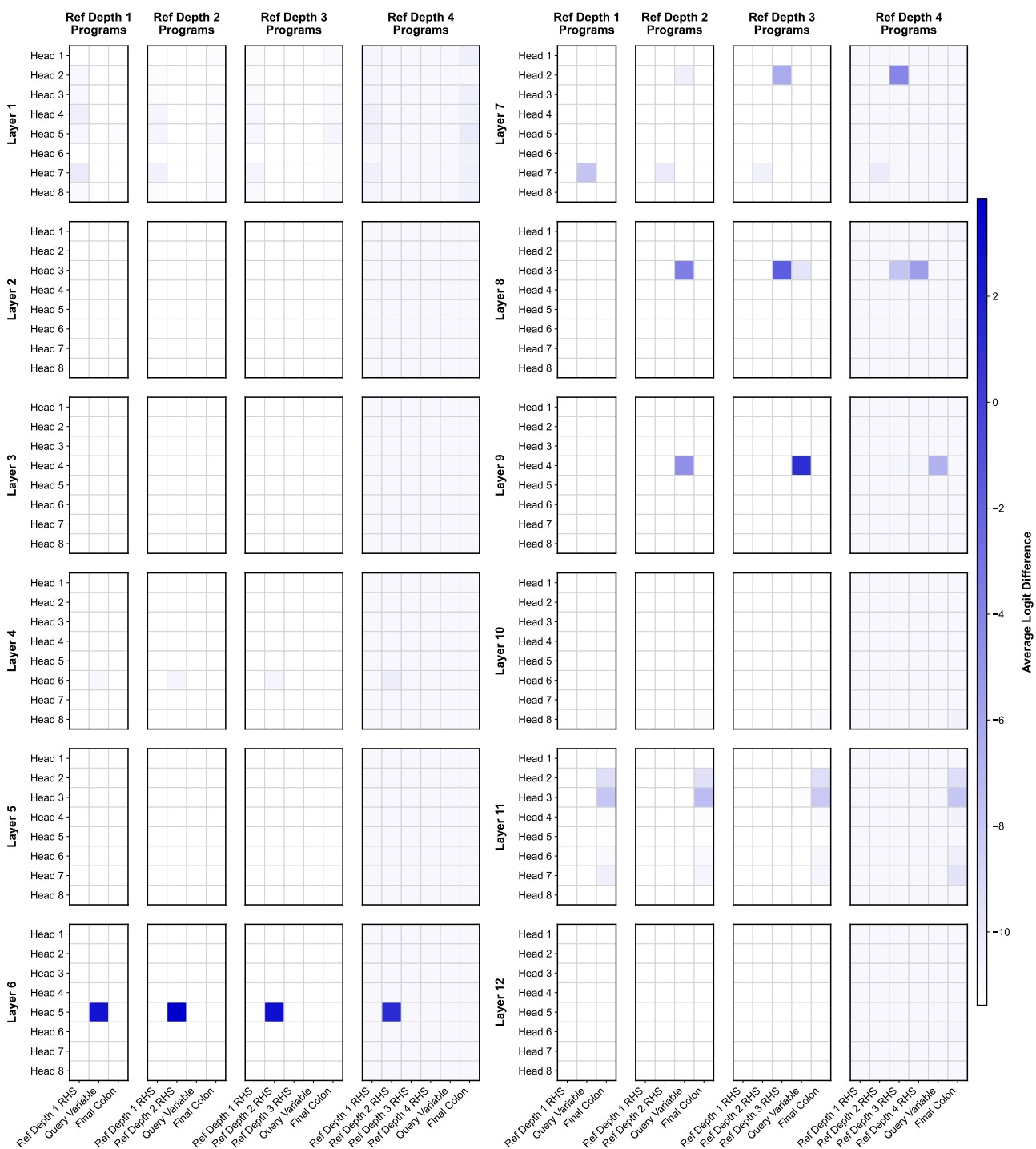

*Figure 10.* Patching results targeting individual attention heads across all 12 layers of the model. For each (head, token) position, we replace only that head's contribution to the residual stream with its counterfactual value and compute logits.

