# OpenReview forum: "How Do Transformers Learn Variable Binding in Symbolic Programs?"
_ICML.cc/2025/Conference — ICML 2025 poster_

### Official Review · Reviewer_ZyPs · 2025-03-12

**Overall Recommendation:** 3

**Summary:**

This paper studies how transformers can learn to implement variable binding through training. In particular, the authors focus on a specific task where the transformer is given a synthetic program where variables are assigned either numerical values or other variables, and the target is to correctly return the numerical value corresponding to a queried variable. By closely inspecting and probing the training trajectories, the authors identify three distinct phases on how transformers learn to implement variable binding. Initially, it simply recognizes the desired output should be a numerical value, then, it leans to use heuristics and shortcuts to generate correct output occasionally, and finally, it leans a systematic mechanism and produces nearly perfect performance over the test set. The study offers some interesting observations on how transformers can be trained to acquire capabilities involving manipulating and tracing symbols.

**Claims And Evidence:**

Yes, the authors have carefully designed the experiments to empirically validate their claims and hypotheses.

**Essential References Not Discussed:**

I am not aware of important references that are not discussed.

**Experimental Designs Or Analyses:**

Overall, the experimental design is good. My major concern is on the choice of the test set. Since the authors claim that at the end of training, there is "emergence of a systematic mechanism for dereferencing", it is important to have a thorough test on whether the nearly perfect performance is indeed due to learning a systematic mechanism, rather than being just memorization (since the test set and the training set use the same set of variable names and numerical values, and the program is highly structured).

**Methods And Evaluation Criteria:**

Although the task is simple and synthetic, it allows to verify the claims of the paper in a controlled way. However, in addition to the current empirical setting, I think that the authors should really consider a test set where the program structure remains the same as the training set, but the variables and the numerical values differ from the training set. Since the underlying logic to implement variable binding does not restrict to a fixed set of variables names or numerical values, in order to claim that the transformer "leaned a systematic mechanism for variable binding", it is necessary to demonstrate that the model is able to achieve nearly perfect performance on a test set where neither variable names nor numerical values are seen during training. For example, the authors may consider splitting the 26 letters into 13 for training and 13 for testing, and additionally choose integers from 11 to 20 for the numerical values in the test set.

## update after rebuttal

During the rebuttal, the authors provided a suite of new empirical results to further verify their claims in the paper. The new experiments cover 3 distinct out-of-distribution settings and, in my opinion, substantially improve the quality of the paper. I updated my initial rating based on the new results. Overall, I think this is a good work and I'd like to recommend acceptance.

**Other Comments Or Suggestions:**

I see occurrence of "fig. 2a", "Fig. 2d", "Figure 3(a)". Please be consistent with naming and referencing.

**Other Strengths And Weaknesses:**

The clarity of the paper can be greatly improved. Definitions and technical terminology should be properly stated. The authors should provide details on how logits are computed at each layer in Figure 3. In the caption of Figure 3(c), there is a missing reference to Appendix which is now read as "shown in Appendix TODO".

**Questions For Authors:**

I would be very interested to know what happens if the variables names and numerical values in the test set differ from those in the training set.

**Relation To Broader Scientific Literature:**

Understanding how transformers can learn to manipulate symbols and perform basic reasoning tasks is very important. Although the distinct learning phases are not very surprising, given that in some other cases it has also been shown that the learning curve of transformers demonstrates multiple jumps followed by long plateaus, the detailed causal analysis in this paper is interesting.

**Theoretical Claims:**

NA (There is no theoretical claim that requires a proof in the paper)

---

> ### Author Rebuttal · Authors · 2025-04-01
>
> Thank you for your assessment that we "carefully designed the experiments to empirically validate [our] claims" and for your thoughtful suggestions. Your questions about generalization inspired several new experiments that strengthen our findings.
>
> ## Does the model actually learn a systematic mechanism?
>
> We appreciate the critical perspective on whether the model indeed learns a systematic mechanism in addition to a shallow heuristic, and conduct generalization experiments to further investigate the matter. Since our model is trained from scratch rather than pre-trained, testing on completely unseen tokens would mean testing on untrained embeddings, which wouldn't be meaningful. Instead, we conducted three rigorous generalization experiments:
>
> - **Held-out variable/number combinations**: We trained a model where specific variable/number combinations, randomly sampled and constituting ~10% of the original training data, were excluded from training but included in testing. The model successfully generalized to these unseen combinations, demonstrating compositional generalization rather than memorization. See [the compositional generalization plot](https://imgur.com/a/comparison-of-original-model-vs-compositional-generalization-model-qq9Z8wF) for further details. (direct link to new plot: https://i.imgur.com/esnjS3O.png)
>
> - **Program length generalization**: Testing on programs of 2-25 lines of assignment (vs. 16 in training), we found excellent generalization on programs where the answer is not on the first line, however when the answer is on the first or second line and the first line heuristic is used we see poor performance on shorter and longer programs. See [the program length generalization plot](https://imgur.com/a/model-accuracy-by-program-length-PPHwEyT) for further details (direct link to new plot: https://i.imgur.com/dIaCWx8.png)
>
> - **Chain depth generalization**: Testing on programs with 1-13 hops (vs. max 4 in training), we again observed near-perfect generalization, even for 13-hop programs despite our model having only 12 layers. We again see that when the answer is on the first or second line and the first line heuristic the model is worse at higher depth programs. See [the chain depth generalization plot](https://imgur.com/a/model-accuracy-by-number-of-hops-KZytngj) for further details. (direct link to new plot: https://i.imgur.com/C2XEVwc.png)
>
> In the paper, we claim that the model uses a shallow heuristic for programs where the answer is in the first or second line, and uses a systematic solution for other programs.
>
> When the answer isn’t on the first or second line, the model generalizes far beyond the structures seen during training – to unseen variable/constant combinations, deeper chains, and longer programs. These results further support our claim that the model learns a "systematic mechanism for dereferencing" rather than memorizing patterns.
>
> Moreover, when the answer is on the first or second line, we see seriously degraded generalization performance. These results further support our claim that these are exactly the inputs that a shallow heuristic is used to solve the task.
>
> **Action:** We will add these new experiments and plots.
>
> ## Addressing clarity issues:
> - We will provide clearer definitions of technical terminology
> - We will add details on how logits are computed at each layer in Figure 3
> - We will fix the missing reference in Figure 3(c)
> - We will ensure consistent naming and referencing throughout (e.g., "fig. 2a" vs "Fig. 2d")
>
> We will improve the readability of all figures by:
> - Increasing font sizes
> - Adding clearer labels and legends
> - Ensuring consistent formatting
>
> **Action:** We will implement all these improvements in the camera-ready version.
>
> **Thank you for your valuable suggestions that led to these additional experiments! Do these results and planned improvements address your concerns about whether we've demonstrated a truly systematic mechanism?**

---

### Official Review · Reviewer_eScW · 2025-03-13

**Overall Recommendation:** 3

**Summary:**

In this paper, the "variable binding" ability of Transformer model is studied, which is to autonomously assign correct values to symbolic variables. The paper focuses on controlled experimental design. A symbolic program dataset is constructed, which consists of programs that involve value passing among variables. Based on the learning results on this dataset, it is concluded that the Transformer model has strong variable binding ability without specific training.

**Claims And Evidence:**

In my view, the claims are not well-supported by the evidence: 1) The constructed dataset is too limited. 2) The experimental results lack deep insights. More details are discussed in the experimental design part below.

**Essential References Not Discussed:**

NA

**Experimental Designs Or Analyses:**

1. The construction of the datasets is limited. The programs involves only number assignments among variables, which is a very special type of symbols. It would be better to introduce broader type of symbols (e.g. a is left to b, b is left to c, then a is left to c) and more complicated program structures. In my view, this is not difficult since variable assignment is common in programming tasks.

2. The experimental results are somehow superficial. In the current experiments, the values and symbols are treated equally as tokens. From the results, it is hard to tell whether the model distinguishes between them and actually learns what is a symbol or a number. In my view, we can also explain the experimental results from simply the view of attention weights. After training, the Transformer model just associate the symbols based on attention weights according to their common appearance in the dataset. From this view point, the results do not reflect insight on symbolic perspective. I think the current experiments lack strong evidence to eliminate this explanation.

**Methods And Evaluation Criteria:**

Even though the constructed programs are interesting, I think that the dataset design is still limited to justify the results. More details are discussed in the experimental design part below.

**Other Comments Or Suggestions:**

- The writing can be significantly improved. The paper is hard to read due to the lacking of clarity. Furthermore, the paper needs further proofreading. There are a number of typos in the paper, such as:

  - Line 129: 90% training, 0.2% validation, and 0.98% testing.

  - Line 270: Appendix TODO

- The visual effect in Fig. 2d needs refinement.

**Other Strengths And Weaknesses:**

Strengths:

- I think the paper proposes an interesting paradigm in analyzing the symbolic learning ability of Transformer.

Weaknesses:

- The experimental design is not sufficient to support the major arguments as discussed above.

**Questions For Authors:**

- How three phases are decomposed? Are they just three stages of decreasing the learning rate of SGD optimization, or naturally appeared in the experiments without external parameter changing?

- What objective is used for training the Transformer? Is it auto-regressive next token prediction?

- How to explain Fig. 2c? I wonder why 1-hop accuracy converged slower than those with more hops.

**Relation To Broader Scientific Literature:**

The paper is related to the researches on understanding the principles of Transformer models. Even though the purpose of the paper is to study symbol bindings, in my view, the results are similar to analyzing the attention weights among tokens in previous researches.

**Theoretical Claims:**

No theoretical results are included in the paper.

---

> ### Author Rebuttal · Authors · 2025-04-01
>
> Thank you for your feedback that our paper "proposes an interesting paradigm in analyzing the symbolic learning ability of Transformer." We appreciate your critical assessment and have conducted new experiments to address your concerns.
>
> ## Addressing key concerns:
> ### 1. "The model does not distinguish between symbols and values"
>
> We conducted new subspace analysis experiments specifically to address this concern:
>
> 1. We applied PCA to the Transformer's residual stream vectors for the right-hand-side tokens of program lines across many different inputs.
> 2. We trained linear classifiers (with L1 regularization) identify principal components of the residual stream correlated with numerical constants (10 components) and variable names (26 components), respectively.
> 3. We performed interchange interventions on these selected components, using counterfactual examples that target the numerical constant or the variable name, achieving high success rates for both numerical constants (92.17%) and variable names (87.08%).
>
> **New findings:** The 2D UMAP projections of these subspaces across three training checkpoints (steps 1200, 17600, 105400) reveal a clear evolution toward separated clusters for these token types. By the final checkpoint, the model distinctly represents symbols and values in different subspaces. See the new [the variable-name subspace scatter plot](https://imgur.com/a/2d-umap-projection-of-variable-name-residual-stream-subspace-projection-26-selected-pca-components-KBEft1t) and [the numerical-constant subspace scatter plot](https://imgur.com/a/2d-umap-projection-of-numerical-constant-residual-stream-subspace-projection-26-selected-pca-components-SCyS18B) for more details (direct links to new plots: https://i.imgur.com/ugP0baG.jpeg, https://i.imgur.com/479j8Ch.jpeg).
>
> **Action:** We will add these new experiments and plots.
>
> ### 2. Questions about training
> > How are the three phases decomposed?
>
> The three phases emerge naturally during training without external parameter changes. They represent qualitative shifts in strategy that correspond to sharp transitions in performance (from ~12% to ~56% to >99% accuracy). We use a linear decay learning rate schedule with 750 warm-up steps, but the phase transitions do not align with learning rate schedule changes.
>
> > What objective is used for training the Transformer?
>
> We use standard autoregressive next-token prediction (causal language modeling) as stated in the Appendix section "Training Setup."
>
> **Action:** We will add this information more explicitly in the main text.
>
> > How to explain Fig. 2c? Why 1-hop accuracy converged slower than those with more hops?
>
> This results from our task structure. In 1-hop programs, the answer is always a numerical constant directly assigned to the queried variable. The model's early "line-1/line-2 heuristic" works well for multi-hop programs where the first line often contains the root value, but performs poorly on 1-hop programs where the answer could be on any line.
>
> **Action:** We will explain this in the figure caption.
>
> ### 3. Task design
> While we acknowledge the controlled nature of our task, its simplicity is intentional. By using a minimal design, we can perform detailed causal analysis that would be challenging with more complex symbolic tasks. Our new generalization experiments show the model learns a truly general mechanism rather than simply memorizing patterns.
>
> We would like to clarify an important aspect of our experimental design: our model is trained entirely from scratch without any pre-training on natural language or programming tasks. When the model first encounters our variable assignment syntax (e.g., "a=1"), it has no prior knowledge of what these symbols mean or how they relate to each other. From this perspective, using alternative symbolic relationships like "a is left of b" would be functionally equivalent - the model would still need to learn these relationships from scratch.
>
> Our focus was on creating a minimal yet challenging test bed where we can precisely analyze how variable binding mechanisms emerge during training. The apparent simplicity of our task masks its fundamental complexity: the model must still develop sophisticated capabilities to track multi-step variable dependencies while ignoring distractors, regardless of the specific symbolic relationship used.
>
> The controlled design was crucial for our mechanistic interpretability goals, allowing us to isolate and analyze the precise causal pathways through which variable binding is implemented in the network.
>
> ## Additional improvements:
> - We will fix all typos, including the dataset split percentages
> - We will improve the clarity of Fig. 2d
> - We will add clearer explanations of the training objective and methodology
>
> **Action:** We will implement all these improvements in the camera-ready version.
>
> **Thanks again for your helpful feedback! Do these clarifications and planned changes address your concerns?**

---

> > ### Comment · Reviewer_eScW · 2025-04-04
> >
> > Thanks for the detailed responses and the additional experimental results. I appreciate the additional information, as well as the explanation on conducting a "minimalist" test for justifying the target claims. While I still feel that the results don't justify that the Transformer can conduct complex symbolic "reasoning", since symbol binding itself is not quite different from build correlations through attention weights.  Overall, I will take the responses into consideration for my final evaluation.

---

### Official Review · Reviewer_kEn5 · 2025-03-13

**Overall Recommendation:** 3

**Summary:**

The authors investigate how variable binding is learned by decoder-only transformer models. In particular, they use causal interventions to understand how transformers propagate binding in tasks where they are asked for the value of a variable, e.g., "c" where ("c=b, p=f, b=a, f=2, a=1") is given. Their key findings are: 1) A general mechanism for dereferencing is acquired in later stages of training, and gradually subsumes heuristics the model learned early on, 2) the full program state is not stored in the residual stream, and 3) certain attention heads' role in routing to dereference can be understood. Finding #1 is particularly interesting, as previous narratives around the acquisition of capabilities in transformers hypothesized that these replace early heuristics and are learned alongside, rather than on top of, them.

The paper is well-written and clear, and a very nice interactive webpage is provided alongside the paper where results are clearly explained and can be explored.

**Claims And Evidence:**

Claims are well-supported.

**Essential References Not Discussed:**

None

**Experimental Designs Or Analyses:**

Experiments are suitable.

**Methods And Evaluation Criteria:**

They used the existing method of interchange interventions which is a sufficient patching method to isolate heads relevant to the mechanism they are studying.

**Other Comments Or Suggestions:**

* Figure 4 is very dense; would prefer large legend at the top, and perhaps thicker lines. Some of the color differences are not that substantial.
* Figure 3 - text very small again. Maybe there is a way you can label ticks with an example as reference to make it easier to parse; e.g., say for example "x=3:" and then label x, =, 3, :.
* Line 95 column 2 - "open question of how does..." incorrect grammar.
* Line 136 column 2 - "hand side of the quality" → "equality".
* Line 271 - there is a "TODO".
* Line 291 column 2 - "this is evidence that a uniform..." incorrect grammar, and not sure this is a completely valid claim. Representations may be similar (not uniform), but the attention head being implicated on both is a function of the representation after q,k projections, not residual stream? Though I concede likely to be similar.
* Both in the intro and conclusion you make reference to the ongoing symbolic vs. connectionist debate and claim to contribute by showing that models can solve such tasks, but models have long been shown to complete vastly more complex tasks that require handling complex inter-dependencies, so this may be overblown.

**Other Strengths And Weaknesses:**

None

**Questions For Authors:**

Why do you think linear probing works so poorly yet you are able to see clean effects from patching?

**Relation To Broader Scientific Literature:**

The paper presents various results around variable binding in transformers. The most interesting of these by far is the way in which the relevant circuits form, with the behaviour itself being fairly uninteresting. As such, the paper forms a reasonable contribution to the literature.

**Theoretical Claims:**

No theoretical claims are made in the body of the paper.

---

> ### Author Rebuttal · Authors · 2025-04-01
>
> Thank you for your positive assessment that our paper is "well-written and clear" with claims that are "well-supported." We're especially pleased you highlighted our most interesting finding: that relevant circuits form with "behavior itself being fairly uninteresting."
>
> ## Response to your question: Linear probing vs. patching
> > Why do you think linear probing works so poorly yet you are able to see clean effects from patching?
>
> Our linear probing experiment attempts to extract a complete program state from a single vector, while our patching experiments reveal the causal path of specific information through the network. The latter succeeds because the model implements variable binding as a dynamic process of information routing rather than as static state representations.
>
> The poor linear probing results suggest the model doesn't maintain a complete program state in a linearly decodable format. Instead, our patching results reveal that the model learns a more efficient strategy that dynamically tracks only the relevant variable bindings through specialized attention patterns.
>
> **Action:** We will make this more explicit.
>
> ## Addressing your other feedback:
> - **Grammar issues:** We will correct all grammatical errors, including:
>     - Line 95: "open question of how does" → "open question of how"
>     - Line 136: "hand side of the quality" → "equality"
>     - Line 291: We'll rephrase the statement about uniform representations
>
> - **Figure density:** We will redesign Figure 4 with:
>     - A larger, clearer legend at the top
>     - Thicker, more visually distinct lines
>     - Consistent labeling schemes
>
> **Action:** We will implement all these improvements in the camera-ready version.
>
> **Thanks again for your helpful feedback! Do these clarifications and planned changes address your concerns?**

---

### Official Review · Reviewer_stBZ · 2025-03-14

**Overall Recommendation:** 4

**Summary:**

This paper performs a mechanistic interpretability study on a transformer trained on a synthetic task that requires tracking values assigned to variables in a mock programming language. Programs consist of 16 variable assignments, and the transformer has 12 layers. The authors use interchange interventions to identify the path through the transformer that the original value of the variable is transmitted through. Given a variable whose value is queried at the end of the sequence, the transformer solves the task by attending to the last assignment to that variable, doing so across multiple layers if necessary to figure out the chain of assignments and identify the constant used for the first assignment. This is intermixed with naive solutions that are specific to the first and second lines. The authors analyze the evolution of this solution over training and identify three main phases.

**Claims And Evidence:**

Yes, for the most part.

**Essential References Not Discussed:**

Not that I'm aware of.

**Experimental Designs Or Analyses:**

Yes, although I did not check the appendix.

**Methods And Evaluation Criteria:**

Yes.

**Other Comments Or Suggestions:**

1. 155 right: this is redundant
2. Fib 2b is too hard to read. The colors and numbers should be related visually, perhaps with different shades of gray.
3. Fig 4: It's hard to keep track of the line colors.

**Other Strengths And Weaknesses:**

**Strengths**

1. The paper is straightforward and generally well-written.
2. The experiments essentially validate the authors' claims convincingly in terms of how information flows through the transformer.
3. The phases discovered during training are interesting.
4. They made a snazzy website to view the results.

**Weaknesses**

1. The scope of the paper is limited to a single transformer trained on a simple synthetic task, so its real-world impact is limited.
2. The synthetic task is simple and could have been made more complicated. See Questions.
3. Some of the figures are hard to read.

**Questions For Authors:**

1. Why exactly 17 lines for all examples? Did you try testing on longer or shorter programs to check if the construction it learns is sensitive to the length of the training data?
2. It seems like the learned solution is limited by the number of layers in the transformer. What happens when you test on any examples with chains longer than 12?
3. Did you try reproducing this over multiple training runs?
4. This analysis is likely sensitive to the distribution of the training data. Did you try other distributions?
5. Fig 3a: Why is logit difference on a scale from 0 to 1? Even if it's probabilities, shouldn't it be on a scale from -1 to 1?

**Relation To Broader Scientific Literature:**

This is a mechanistic interpretability paper on a new synthetic variable assignment task.

**Theoretical Claims:**

This paper does not have theoretical claims.

---

> ### Author Rebuttal · Authors · 2025-04-01
>
> We thank you for your thorough review and positive feedback that our paper is "straightforward and generally well-written" with experiments that "validate the authors' claims convincingly."
>
> ## Responses to your questions:
>
> ### Q1&Q2: Why exactly 17 lines? Did you try testing on longer/shorter programs? What happens with chains longer than 12?
> See the response to Reviewer ZyPs for the results of new generalization experiments.
>
> ### Q3: Did you try reproducing this over multiple training runs?
> Yes, we trained additional models with different random seeds. All runs exhibit the same three distinct learning phases with similar transition points, demonstrating the robustness of our findings. See [the test set accuracy plot across different training run seeds](https://imgur.com/a/comparison-of-test-set-accuracy-across-different-seeds-OXfXTSF) for more details (direct link to new plot: https://i.imgur.com/cvf0vzO.png).
>
> **Action:** We will add this new experiment and plot.
>
> ### Q4: Did you try other distributions?
> Yes, we explored different sampling strategies during task design. Our final distribution was deliberately constructed to be as challenging as possible while remaining learnable.
>
> Simpler distributions (e.g., with fewer distractor chains, uniform sampling of variables, or shorter chains) made the task trivially solvable using surface-level heuristics. For instance, without our weighted sampling approach (where we choose chains to extend with probability proportional to chain length cubed), the model could simply learn to associate the answer with the longest variable chain.
>
> Our rejection sampling procedure ensures balance across referential depths while maintaining sufficient distractor chains that branch from the main query chain. This forces the model to genuinely track variable bindings rather than rely on pattern matching.
>
> This challenging distribution was essential for studying true variable binding mechanisms rather than shortcut learning.
>
> **Action:** We will make this more explicit in the appendix.
>
> ### Minor points
> The logit differences were normalized to a [0,1] scale for visualization clarity. We agree this could be clearer and will explicitly state the normalization in the figure caption.
>
> We acknowledge your concern about figure readability. In the camera-ready version, we will:
>
> - Increase font sizes throughout all figures
> - Use more visually distinct colors and line styles in Fig. 2b and Fig. 4
> - Add clearer legends and improve visual mapping between colors and values
>
> **Thanks again for your helpful feedback! Do these clarifications and planned changes address your concerns?**

---

> > ### Comment · Reviewer_stBZ · 2025-04-08
> >
> > Thank you for your responses. Most of my concerns have been addressed. I'll update my score accordingly.
> >
> > I just have one unresolved question: What if the number of variable hops is much larger than the number of layers, like 50 hops and 12 layers? I don't think 13 hops and 12 layers is enough to test my original concern.

---

### Decision · Program_Chairs · 2025-05-01

**Decision:**

Accept (poster)

**Comment:**

This paper studies variable binding in Transformer neural network on a synthethic task that requires tracking values assigned to variables in a mock programming language. Leveraging tools from mechanistic interpretability, it is shown how the capacity for variable binding forms in three key stages, and what role various parts of the Transformer play for implementing variable binding.

This paper initially received mixed reviews. While reviewers are largely positive about the presentation, the quality of the experiments, the validity of the claims and the overall contribution, there are a few limitations worth highlighting. In particular, only a single configuration of the Transformer is studied, and only a single task is considered, which affects the generality of the claims made. Reviewers pointed out how "the values and symbols are treated equally as tokens", which leaves some doubt whether the model really treats them differently, and how it is necessary to compare on an out-of-distribution test-test. Both of these concerns were reasonably well addressed during the rebuttal.

Following the reviewer discussion, all reviewers are in agreement that this work should be accepted, and two reviewers increased their scores. The AC agrees that this paper is of sufficient quality and that there is enough of a contribution to accept. That said, I would ask the authors to adjust their title in light of the feedback received to better reflect the contribution of the paper, including its limitations.